# MW Synthesis of ZIF-7. The Effect of Solvent on Particle Size and Hydrogen Sorption Properties

**Vladimir A. Polyakov \*** , **Vera V. Butova \*** , **Elena A. Erofeeva, Andrei A. Tereshchenko and Alexander V. Soldatov**

The Smart Materials Research Institute, Southern Federal University, Sladkova 178/24, Rostov-on-Don 344090, Russia; Bulanova@sfedu.ru (E.A.E.); antereshenko@sfedu.ru (A.A.T.); soldatov@sfedu.ru (A.V.S.)

\* Correspondence: vlpolyakov@sfedu.ru (V.A.P.); vbutova@sfedu.ru (V.V.B.)

**Abstract:** We report here fast (15 min) microwave-assisted solvothermal synthesis of zeolitic imidazolate framework material (ZIF-7). We have optimized solvent composition to achieve high porosity and hydrogen capacity and narrow particle size distribution. It was shown that synthesis in N,N-diethylformamide (DEF) results in a layered ZIF-7 III phase, while N,N-dimethylformamide (DMF) as solvent leads to a pure ZIF-7 phase in microwave conditions. A mixture of toluene with DMF allows the production of pure ZIF-7 material only with the triethylamine additive. Obtained materials were comprehensively characterized. We have pointed out that both X-ray diffraction and infrared spectroscopy could be used for the identification of ZIF-7 or ZIF-7 III phases. Although samples obtained in DMF, and in a mixture of DMF, toluene, and triethylamine were assigned to the pure ZIF-7 phase, solvent composition significantly affected the size of particles in the material and nitrogen and hydrogen adsorption process.

**Keywords:** ZIF-7; hydrogen storage; microwave synthesis

## 1. Introduction

The problem of depletion of natural energy sources, such as natural gas, oil, coal, as well as the continuing growth in energy consumption in the 21st century, creates a colossal problem in the future. Humanity needs a transition to clean renewable energy sources. Solar and wind energy meet these criteria, but they are also characterized by frequent interruptions in energy production.

One of the most promising candidates for the role of a new energy source is hydrogen due to its enormous energy density by weight of 142 MJ/kg, which is about three times the energy released by gasoline and seven times the energy released by coal [1–4]. However, it is characterized by a very low specific energy per unit volume. It creates problems related to its storage and transportation [5]. Another crucial step is the purifying of the obtained hydrogen. It could be produced according to the list of protocols such as the water–gas shift with subsequent steam–methane reforming [6], catalytic formic acid dehydrogenation [7], water splitting [8] and others. However, the obtained hydrogen has to be purified or collected from the product mixture before it can be used. For this purpose, membrane-based separators could be applied. One of the possible solutions is the use of nanoporous materials, in particular the metal–organic frameworks (MOFs), for the physical adsorption of hydrogen, which reduces energy costs without requiring low temperatures and high pressures for liquefaction [9–15].

Flexible MOFs have recently attracted the attention of researchers due to their unique structure and their use as gas separators [16,17], molecular probes [18,19], and drug delivery agents [20]. Many MOFs of this type can deform reversibly, changing their geometry and pore size by rotating linkers around sigma bonds when interacting with gas molecules. Such structural deformation can

lead to a sharp increase in the adsorption capacity of the material, as well as to an increase in the sorption selectivity. Thanks to this deformation of the geometry, we can register stepped isotherms of gas adsorption [21–23].

One of these flexible MOFs is the zeolitic imidazolate framework material (ZIF-7) framework containing zinc tetrahedrally coordinated to the nitrogen atoms from benzimidazole molecules as linkers. Such coordination leads to the formation of a sodalite (SOD) topology [24]. At the same time, there are 3 phases of ZIF-7: rhombohedral *R-3m* ZIF-7-I, triclinic *P-1* ZIF-7-II, and the most dense monoclinic *C2/c* ZIF-7-III [25]. Phases ZIF-7-I and ZIF-7-II can reversibly pass into each other when the amount of guest molecules of dimethylformamide (DMF) or other organic solvent changes, as well as when the pressure of the gaseous product changes. However, the transition of ZIF-7-I and ZIF-7-II to the ZIF-7-III phase occurs irreversibly when the pores of the material are saturated with water molecules. This fact can be explained by the process of Zn-N bonds selective hydrolysis. Thus, the method for ZIF-7 synthesis, in particular the choice of solvent, directly influences the process of phase formation, and, therefore, the efficiency of the sorbent generally. The ZIF-7-I phase has the highest sorption capacity, which means that it is the most attractive for various gas applications [25,26]. This material is already being used as a separator [27] and for hydrogen storage [28].

ZIF-7-I (hereinafter ZIF-7) was first obtained by Park et al. [29] by solvothermal reaction between zinc nitrate tetrahydrate and benzimidazole in DMF solution. The process proceeded at 130 °C for 48 h. Materials in studies [25,26,30–35] were also synthesized in a similar way. Variations of this technique with varying synthesis times/temperatures was described [36–39]. Methods are also described that allow the preparation of mixed linkers ZIFs, as well as core-shell structures [24,40]. However, all these methods are lengthy and are characterized by medium product yields. Tu et al. [41] proposed a more convenient method, which significantly reduced the synthesis time, excluding heating of the reaction mixture, and increased the product yield (up to 78%) [42]. The mixture of zinc nitrate hexahydrate and benzimidazole was dissolved in a mixture of DMF/methanol in a ratio of 1/1 and stirred at room temperature for 30 min.

The solvent choice plays an essential role in the formation of ZIF-7 phases. As reported by Reif et al. [43,44], replacing DMF with anhydrous diethylformamide (DEF) leads to the crystallization of the ZIF-11 phase. The water content in DEF is a very important criterion because even small amounts of $H_2O$ (over 0.3 wt.%) lead to product contamination with the ZIF-7-III phase. Replacing the solvent with ethanol leads to the formation of the ZIF-7 phase [45–47]; however, the use of ethanol/toluene mixture results in a pure ZIF-11 phase [28,45,46].

The process direction, the product yield, as well as the particles morphology can be changed by introducing various modulators into the reaction solution. Bases-modulators play the most important role. For example, carrying out the synthesis in a concentrated ammonia water solution leads to the formation of the ZIF-7 phase rather than ZIF-7-III [48]. The use of ammonia in a DMF solution accelerates the ZIF-7 synthesis even at room temperature [49]. Organic bases, in particular, triethylamine (TEA), make it possible to obtain ZIF-7 in good yield in acetonitrile solution [50,51]. Diethylamine works in a similar way [52–54]. Among other used modulators, sodium formate can be distinguished [55,56]. It also promotes the formation of the ZIF-7 layer on the various supports, for example, aluminum oxide [57,58].

The morphology is also controlled by the selection of zinc salts. Cai et al. [53] studied the effect of a metal source and modulators on the ZIF-7 particles formation. It was shown that the use of zinc nitrate leads to the formation of spherical particles. Zinc acetate allowed the formation of rhombic dodecahedral particles. ZIF-7 nanocrystals obtained from $ZnCl_2$ have a shape of hexagonal prisms. This phenomenon is associated with a strong interaction between $Cl^-$ and $Zn^{2+}$, which significantly affects the growth kinetics of ZIF-7 nanocrystals. This fact does not depend on the synthesis method, i.e., ZIF-7 nanocrystals synthesized using traditional or microwave heating have a rod-like shape [59]. Polyhedra and rods were formed only in the presence of diethylamine. Zinc acetate dihydrate also

allows obtaining ZIF-7 in a water/DMF mixture within a couple of minutes at room temperature, which makes this technique the most effective [60].

Other synthesis methods were described, for example, using microwave irradiation [27,61], ultrasonics [62], as well as their combination [63]. However, these methods are rare in the literature, although they are very promising, since they can significantly accelerate the reaction rate, reduce the temperature/synthesis time, and also obtain much smaller ZIF-7 particles.

ZIF-7 contains small pores with hexagonal windows of about 0.3 nm, which is optimal for the separation of hydrogen (0.29 nm) from carbon dioxide (0.33 nm) or nitrogen (0.36 nm) [64]. This material was successfully applied for the separation of hydrogen, and it showed high selectivity [58,64,65]. However, the flexibility of the ZIF-7 structure could lead to phase transitions and a "gate-opening" effect, affecting separation properties. In this work, we focused on the preparation of ZIF-7 by the microwave-assisted solvothermal method using various solvents.

## 2. Materials and Methods

Starting materials zinc nitrate hexahydrate ($Zn(NO_3)_2 \cdot 6H_2O$), benzimidazole ($C_7H_6N_2$, bIm), triethylamine ($C_6H_{15}N$, TEA), N,N-dimethylformamide (($CH_3)_2NCHO$, DMF), N,N-diethylformamide (($C_2H_5)_2NCHO$, DEF), methanol ($CH_3OH$), toluene ($C_7H_8$, PhMe), and isopropanol ($C_3H_7OH$) were purchased from SigmaAldrich. Ultra-pure water (18 MΩ·cm) was obtained from distilled water using SimplicityUV (Millipore).

We applied MW heating to obtain ZIF-7 materials with a fixed zinc source, temperature and duration of synthesis. Simultaneously, we varied the solvent composition and TEA additive to trace their effect on phase formation and the properties of the synthesized compounds.

Zinc nitrate hexahydrate 0.0812 g (0.27 mmol) and bIm 0.086 g (0.73 mmol) were dissolved in 5 mL of DEF (*sample ZIF-7 DEF*) or DMF (*sample ZIF-7 DMF*).

For the synthesis of the other two samples, a mixture of DMF and PhMe was prepared in a volume ratio of 3/1 (D/P solution). Then, zinc nitrate hexahydrate 0.1098 g (0.35 mmol) was dissolved in 2.5 mL of D/P solution. Separately, bIm 0.086 g (0.73 mmol) was dissolved in 2.5 mL of D/P solution (*sample ZIF-7 D/P*). TEA (126 μL) was added to the bIm solution to obtain *sample ZIF-7 D/P TEA*. After that, the solutions of zinc nitrate and bIm were mixed. Molar ratios of precursors are presented in Table 1.

**Table 1.** Molar ratio of precursors and conditions applied for optimization of Zn-ZIF-7 MW synthesis.

| Sample Designation | Precursors | | | | Conditions | | Phase According to XRD |
|---|---|---|---|---|---|---|---|
| | $Zn^{2+}$ | bIm | TEA | Solvent | t, °C | Time, min | |
| ZIF-7 DEF | 1 | 2.7 | 0 | DEF | 140 | 15 | ZIF-7 III |
| ZIF-7 DMF | 1 | 2.7 | 0 | DMF | 140 | 15 | ZIF-7 |
| ZIF-7 D/P | 1 | 2 | 0 | DMF/PhMe | 140 | 15 | ZIF-7 + ZIF-7 III |
| ZIF-7 D/P TEA | 1 | 2 | 2.6 | DMF/PhMe | 140 | 15 | ZIF-7 |

Abbreviations: ZIF-7: zeolitic imidazolate framework material.

All the resulting solutions were then poured into 10 mL glass flasks and placed in a CEM Discover microwave oven. The synthesis was carried out at 140 °C, power set to 75 W, and medium stirring was performed for 15 min. After cooling the solutions to room temperature, the precipitates were separated by centrifugation. We washed them three times with DMF and one time with isopropanol. Then, the samples were dried at 60 °C overnight.

An FEI Tecnai G2 Spirit BioTWIN (FEI, USA) was used to perform transmission electron microscopy (TEM) for imaging of the obtained samples. An accelerating voltage of 80 kV was used. Powder X-ray diffraction (XRD) of samples was performed using a D2 Phaser (Bruker AXS Inc., Karlsruhe, Germany) X-ray diffractometer. Data was collected in 2θ range −5°–90° with a step size of −0.01° using Cu Kα

radiation ($\lambda$ = 1.540562 Å) at 30 kV and 10 mA. We used the Jana2006 software package for the analysis of diffraction patterns [66].

A Bruker Vertex 70 spectrometer was applied for measurement of IR spectra. We used attenuated total reflectance (ATR) geometry with air as the reference. The spectra were collected with a resolution of 1 cm$^{-1}$; 128 scans were made in the range from 5000 to 30 cm$^{-1}$ on an MCT detector and a Bruker Platinum ATR attachment.

Nitrogen adsorption isotherms were recorded at $-196$ °C on Accelerated Surface Area and the porosimetry analyzer ASAP 2020 (Micromeritics). The specific surface area values were calculated according to the Brunauer–Emmett–Teller (BET) model using nitrogen adsorption isotherms. Hydrogen adsorption–desorption isotherms were measured at $-196$ °C as well and used to calculate the H$_2$ capacities of materials. Before both measurements, samples were degassed at 250 °C for 12 h under a dynamic vacuum.

## 3. Results and Discussion

All synthesized samples were highly crystalline materials, according to powder XRD (Figure 1a). All diffraction peaks of the sample ZIF-7 DEF could be attributed to the monoclinic ZIF-7 III phase with space group *C2/c* (no. 15) (see details in SI Table S1, Figure S2). Two other single-phase samples ZIF-7 DMF and ZIF-7 D/P TEA, could be considered pure ZIF-7 phase. All reflections were attributed to the rhombohedral symmetry with space group *R-3* (no. 148) (SI Table S1, Figure S3). Sample ZIF-7 D/P is composed of two phases: ZIF-7 III with an admixture of ZIF-7 (Figure S2). We noticed that lattice constants *b* of samples attributed to the ZIF-7 phase were almost similar regardless of solvent, while for parameters *a,* we observed a clear trend. Samples synthesized in a mixture of DMF and PhMe exhibited greater values of parameter *a* than ones obtained in pure DMF. This could be considered as a sign of higher stability of the open modification of ZIF-7 flexible structure in case of synthesis in a mixture of solvents. A broadening of reflections in the ZIF-7 D/P TEA profile was associated with the small size of particles, which was estimated according to Williamson-Hall analysis as 46 nm (Figure S4).

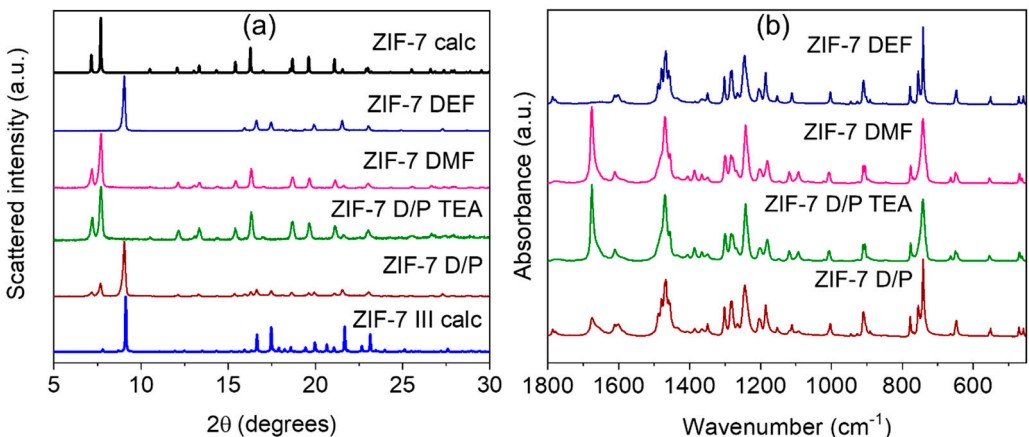

**Figure 1.** Powder X-ray diffraction (XRD) patterns (**a**) and FTIR-spectra (**b**) of synthesized samples. In part (**a**) "ZIF-7 calc" (black line) and "ZIF-7 III calc" profiles (blue line) were calculated according to crystallographic data from [25].

Based on profile analysis, we chose optimal conditions for ZIF-7 MW synthesis as were used for samples ZIF-7 DMF and ZIF-7 D/P TEA. We concluded that for synthesis in the DMF solution, the TEA additive obstructs the formation of the ZIF-7 phase (see SI Figure S5), while for a mixture of DMF and toluene only with TEA additive, a single-phase ZIF-7 sample could be obtained.

FTIR spectra of synthesized samples are provided in Figure 1b. All obtained materials are constructed from zinc ions and benzimidazole linkers. In this way, regardless of the phase composition, samples should exhibit similar FTIR spectra. However, due to significant differences in the crystal

structure of the obtained phases, FTIR spectra also deviate from each other, and they could be used for phase identification as well (Figure S6, Table S2). According to XRD data, samples ZIF-7 DMF and ZIF-7 D/P TEA are single-phase ZIF-7 materials. In good agreement with this statement, their spectra contain the same modes. ZF-7 DEF sample was attributed to the ZIF-7 III phase, and its FTIR spectrum deviates from others. Finally, the ZIF-7 D/P sample is composed of two phases, and its spectrum contains modes assigned to both ZIF-7 and ZIF-7 III phases. The most significant differences could be observed in the benzene ring vibrations. In the ZIF-7 phase, all benzene rings are located inside spherical pores, while in the ZIF-7 III phase, they are directed into the spaces between layers (Figure S1). In more detail, modes in the range 450–665 cm$^{-1}$ could be assigned to vibrations of C-C-C bonds in the benzene ring of the linker. In this region, two features could be observed in the ZIF-7 DEF (ZIF-7 III) spectrum: an extinction of the peak at 665 cm$^{-1}$ and a higher intensity peak at 455 cm$^{-1}$. Out-of-plane vibrations of C-H bonds give rise to peaks in region 740–905 cm$^{-1}$ [67]. This region is quite similar for both phases. The mode at 1005 cm$^{-1}$ is attributed to C-C-C trigonal bending, and it could be observed in all spectra as well. The next region at 1090–1275 cm$^{-1}$ is associated with C-*H in*-plane bending [67]. It contains a list of features. The spectrum of ZIF-7 DEF (ZIF-7 III) exhibits one peak at 1110 cm$^{-1}$, while the spectrum of ZIF-7 phases shows a double-peak in this region: 1090 and 1115 cm$^{-1}$. The next mode at 1155 cm$^{-1}$ can be observed only in the ZIF-7 III phase spectrum. C-C stretching vibrations give rise to a peak at 1240 cm$^{-1}$ in all spectra [67]. The region 1300–1365 cm$^{-1}$ is attributed to C-N stretching vibrations, and it is similar for both ZIF-7 and ZIF-7 III phases. Modes in region 1390–1610 cm$^{-1}$ could be attributed to C=C stretching of a benzene ring. This region differs significantly for spectra of ZIF-7 and ZIF-7 III phases. We observed the extinction of peaks at 1390 and 1405 cm$^{-1}$ in the ZIF-7 III spectrum, while a peak at 1605 cm$^{-1}$ observed for the ZIF-7 phase transforms into a double-peak at 1600 and 1610 cm$^{-1}$ in the ZIF-7 III spectrum. The peak at 1675 cm$^{-1}$ in the spectrum of the ZIF-7 phase is assigned to C=N bonds [68].

We compared the shape of crystals for two single-phase samples ZIF-7 DMF (ZIF-7 phase) and ZIF-7 DEF (ZIF-7 III) (Figure 2). In good agreement with the XRD data, the samples exhibited different shapes of particles. The ZIF-7 DMF sample has a rhombohedral symmetry, and its crystals showed a prismatic shape. In contrast, ZIF-7 DEF crystallized in flat plates about 1.5–2 μm.

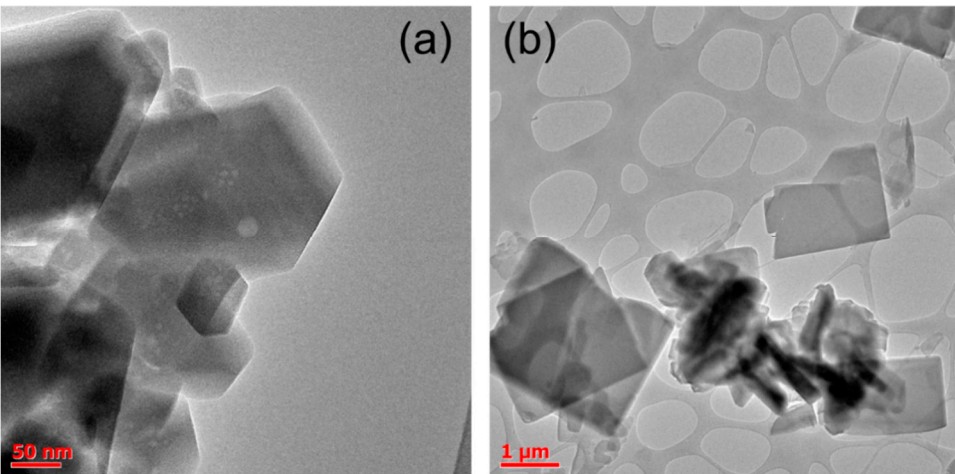

**Figure 2.** Transmission electron microscopy (TEM) images of synthesized samples ZIF-7 N,N-dimethylformamide (DMF) (**a**) and ZIF-7 N,N-diethylformamide (DEF) (**b**).

In good agreement with the XRD data, TEM images revealed the same shape of crystals in samples ZIF-7 DMF and ZIF-7 D/P TEA assigned to the ZIF-7 phase (Figure 3). Despite their size, all particles exhibit regular hexagonal or square shape, indicating high crystallinity of both samples. We observed the broadening of reflections in the XRD profile of the ZIF-7 D/P TEA sample, attributed to the small size of particles (46 nm, according to the Williamson-Hall analysis). Particle size distributions for ZIF-7

DMF and ZIF-7 D/P TEA samples calculated according to TEM images are provided in Figure 3c,d. They are in good agreement with data obtained from XRD profiles. According to the TEM images, the average particle size for the ZIF-7 D/P TEA sample was estimated as 30–60 nm, while the ZIF-7 DMF sample is composed of nanoparticles about 50 nm, with an admixture of bigger particles about 100–300 nm.

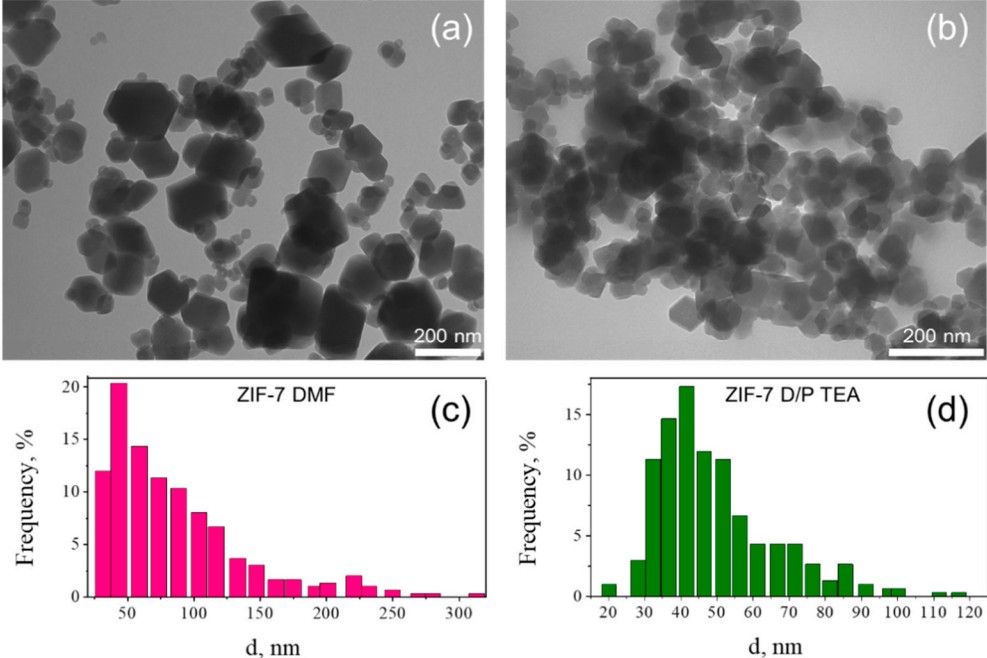

**Figure 3.** Representative TEM images of samples of ZIF-7 DMF (**a**) and ZIF-7 D/P TEA (**b**). Particle size distribution was calculated for 300 particles of ZIF-7 DMF (**c**) and ZIF-7 D/P TEA (**d**) samples. **Abbreviations:** TEA: triethylamine

For two single-phase ZIF-7 samples, nitrogen adsorption-desorption isotherms were measured (Figure 4a). The samples were activated at 250 °C for 12 h under a dynamic vacuum before both measurements. Although ZIF-7 DMF and ZIF-7 D/P TEA have the same crystal structure, according to XRD, their adsorption–desorption isotherms exhibit different shapes. Moreover, the ZIF-7 DMF sample's specific surface area is significantly lower than the ZIF-7 D/P TEA one. Values calculated according to the BET model were estimated as 29 and 284 $m^2/g$ for ZIF-7 DMF and ZIF-7 D/P TEA, respectively. For both ZIF-7 DMF and ZIF-7 D/P TEA samples, nitrogen adsorption is irreversible, with a final amount of nitrogen entrapped inside the pores at low relative pressure. The adsorption branch of the ZIF-7 D/P TEA isotherm contains steps in the low-pressure region attributed to adsorption into the micropores. The pronounced hysteresis loop could be observed at relative pressure higher than 0.7. We assigned it to adsorption into the spaces between nanoparticles in aglomerates. Conversely, the adsorption branch of ZIF-7 DMF isotherm demonstrates nitrogen adsorption blockage. Despite the equivalent phase composition and post-synthetic treatment, pores of the ZIF-7 D/P TEA sample are available for nitrogen molecules, while the ZIF-7 DMF sample can not adsorb them. It could be attributed to the gate-opening effect. As it was reported previously, rotation of benzimidazole linker around σ-bonds leads to the reversible phase transition of ZIF-7. "Closed" modification possesses a pore aperture of about 3 Å, while the "open" phase could permit molecules up to 5 Å [33,69]. In this way, nitrogen could be adsorbed only by "open" modification, while closed pores will be completely unavailable for $N_2$ molecules. It was reported that $CO_2$ or $CH_4$ could initiate ZIF-7 phase transition due to interaction with benzimidazole linkers, while inert $N_2$ will be completely unavailable [33]. However, we have observed that this statement is in good agreement with data for the sample ZIF-7 DMF, while the sample ZIF-7 D/P TEA exhibited obvious permanence for nitrogen molecules and, therefore,

gate-opening effect. As it was reported previously, post-synthetic treatment dramatically affects the shape of nitrogen adsorption isotherms and consequently calculated porosity of ZIF-7 samples [42]. We suppose that solvent exchange is responsible for different sorption processes of ZIF-7 DMF and ZIF-7 D/P TEA samples. According to XRD and TEM data, ZIF-7 D/P TEA sample is composed of nanoparticles about 50 nm, while sample ZIF-7 DMF contains big particles about 90–100 nm along with small ones (40–50 nm). The solvent exchange process occurs faster in small particles, so the ZIF-7 D/P TEA sample could be exchanged faster as well. Secondly, according to synthesis conditions, before solvent exchange and activation, pores of the ZIF-7 DMF sample were filled with pure DMF, while pores of the ZIF-7 D/P TEA sample contained a mixture of DMF, PhMe, and TEA. This mixture could be evacuated in mild conditions due to lower boiling point and chemical activity.

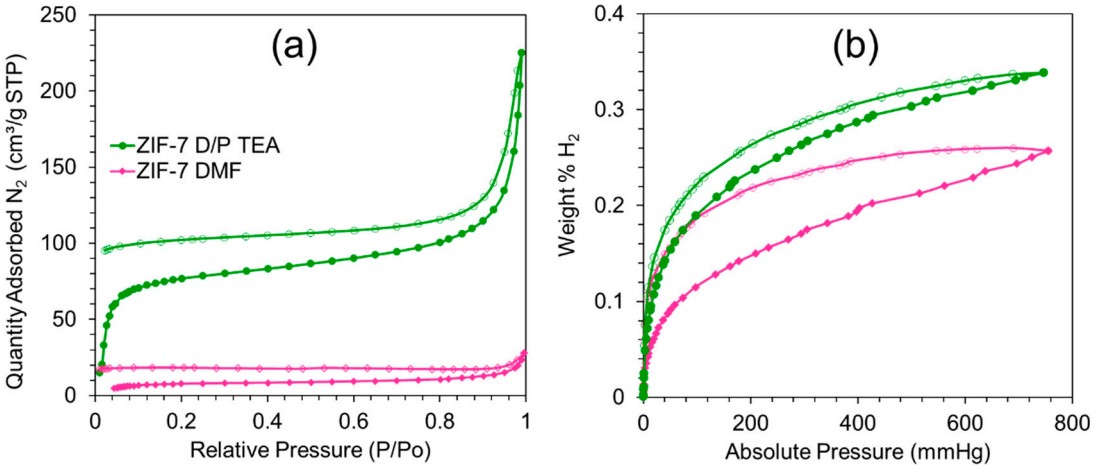

**Figure 4.** Nitrogen (**a**) and hydrogen (**b**) adsorption isotherms of samples ZIF-7 DMF (the pink one with diamond markers) and ZIF-7 D/P TEA (the green one with circle markers). Filled markers represent adsorption branches of isotherms, while empty ones point to desorption branches.

According to nitrogen adsorption isotherms, we have concluded that the "gate-opening" effect is responsible for different porosity of ZIF-7 samples. In the case of ZIF-7 D/P TEA we have observed phase transition to the large-pore modification of ZIF-7, while in the case of ZIF-7 DMF in all pressure range, ZIF-7 preserves the closed-pore structure and it obstructs $N_2$ sorption. However, one could suggest another explanation for this phenomenon: the ZIF-7 DMF sample is not porous at all because the guest molecules or defects somehow block its pores. To exclude this possibility, we additionally measured hydrogen adsorption-desorption isotherms for both samples (Figure 4b). Sample ZIF-7 D/P TEA exhibits higher $H_2$ capacity both at low and ambient pressures. At 20 mmHg, the ZIF-7 DMF sample adsorbed 0.065 wt.% of $H_2$, while the ZIF-7 D/P TEA sample adsorbed 0.11 wt.% of $H_2$ at the same pressure. At 700 mmHg, capacities of $H_2$ were estimated as 0.245 and 0.335 wt.% for samples ZIF-7 DMF and ZIF-7 D/P TEA, respectively. Adsorption and desorption branches of $H_2$-isotherms do no merge with each other. This hysteresis could indicate the interaction of hydrogen molecules with the pore surface. According to previous calculations, $H_2$ in the ZIF-7 structure could be attracted by the two benzene rings of bIm in a perpendicular orientation [70].

In this way, we have observed that despite completely blocked nitrogen adsorption ZIF-7 DMF sample still could adsorb hydrogen molecules. Small $H_2$ molecules are able to penetrate the pores even across close windows. It proves that a solvent could affect phase transition during the activation process of ZIF-7. The sample obtained in DMF solution did not demonstrate a "gate-opening" effect during nitrogen adsorption, while synthesis in a mixture of DMF with PhMe resulted in the material which could adsorb nitrogen and exhibited a relatively high specific surface area.

## 4. Conclusions

We have investigated the effect of solvent and TEA additive on phase formation during the microwave-assisted solvothermal synthesis of ZIF-7. Synthesis in DEF leads to the formation of pure layered ZIF-7 III phase, while a mixture of DMF and toluene results in the formation of a mixture of two phases ZIF-7 III and ZIF-7. TEA additive is crucial for the production of single-phase ZIF-7 material in a mixture of DMF and toluene. Alternatively, ZIF-7 could be synthesized in DMF without additives. To sum up, we have obtained single-phase ZIF-7 material using MW heating for 15 min at 140 °C in pure DMF (sample ZIF-7 DMF) or in a mixture of DMF with toluene and TEA (sample ZIF-7 D/P TEA). Then, we compared ZIF-7 materials obtained according to these optimal conditions. Although both samples exhibited similar XRD patterns and FTIR spectra, they differed from each other significantly. First, sample ZIF-7 D/P TEA showed narrow particle size distribution with an average size of particles about 40–50 nm according to both TEM images and Williamson-Hall analysis of the XRD profile. Sample ZIF-7 DMF contained both nanoparticles about 50 nm and an admixture of bigger crystals about 100–300 nm. Secondly, these samples have demonstrated different flexibility of the structure. Nitrogen adsorption in the ZIF-7 DMF sample was almost completely blocked, while sample ZIF-7 D/P TEA possessed a relatively high specific surface area of 284 $m^2$/g. It indicated that the "gate-opening" effect in the first sample is obstructed, while the second one is flexible and could be transformed into the "open" modification even in a nitrogen atmosphere. Sample ZIF-7 D/P TEA showed slightly higher hydrogen capacity over the whole relative pressure range than the ZIF-7 DMF sample. So sample ZIF-7 D/P TEA is available for both nitrogen and hydrogen molecules. In the case of the ZIF-7 DMF sample, only hydrogen molecules could penetrate pore windows, while $N_2$ molecules are too big, and its adsorption is completely blocked. In this way, we conclude that the solvent affected the phase composition of the samples, particle size distribution, and flexibility of the structure.

**Supplementary Materials:** The following are available online at http://www.mdpi.com/1996-1073/13/23/6306/s1, Content: Crystal structure of ZIF-7 and ZIF-7 III phases, Profile analysis data, FTIR analysis, Nitrogen adsorption. Figure S1: Schematic representation of ZIF-7 (a) and ZIF-7 III (b) unit cells. Blue tetrahedra represent Zn coordination with nitrogen atoms, and gray–carbon atoms. Crystallographic ages are provided on the cell edges., Figure S2: Powder XRD patterns of samples of ZIF-7 DEF (left one) and ZIF-7 D/P (right one): observed (black), calculated (magenta), and their difference (dark blue). The short vertical bars indicate the Bragg positions of the reflections (obtained using the Jana2006 program package)., Figure S3: Powder XRD patterns of samples of ZIF-7 DMF (left one) and ZIF-7 D/P TEA (right one): observed (black), calculated (magenta), and their difference (dark blue). The short vertical bars indicate the Bragg positions of the reflections (obtained using the Jana2006 program package)., Figure S4: Powder XRD patterns of samples ZIF-7 DMF (pink one) and ZIF-7 DMF TEA (green one). Powder profile ZIF-7 calc was calculated according to crystallographic data., Figure S5: Plot of the Williamson-Hall analysis of the profile of the ZIF-7 D/P TEA sample., Figure S6: FTIR-spectra of samples ZIF-7 DEF (dark blue, ZIF-7 III phase) and ZIF- DMF (red, ZIF-7 phase)., Table S1: Details of the profile analysis of the synthesized samples. Table S2. Assignments of observed peaks on experimental FTIR spectra of ZIF-7 DEF (designated as ZIF-7 III) and ZIF-7-DMF (designated as ZIF-7).

**Author Contributions:** Conceptualization, V.V.B.; methodology, V.A.P. and E.A.E.; validation, V.V.B., and V.A.P.; formal analysis, V.V.B. and A.A.T.; investigation, V.A.P., E.A.E. and A.A.T.; writing—original draft preparation, V.V.B. and V.A.P.; writing—review and editing, V.V.B.; visualization, V.V.B. and V.A.P.; supervision, A.V.S.; project administration, A.V.S.; funding acquisition, A.V.S. All authors have read and agreed to the published version of the manuscript.

**Funding:** This research was funded by the Ministry of Science and Higher Education of the Russian Federation, grant number 0852–2020-0019.

**Acknowledgments:** The research was financially supported by the Ministry of Science and Higher Education of the Russian Federation (State assignment in the field of scientific activity, NO. 0852–2020-0019).

**Conflicts of Interest:** The authors declare no conflict of interest.

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
