# Peer review of "MW Synthesis of ZIF-7. The Effect of Solvent on Particle Size and Hydrogen Sorption Properties"

_energies, doi:10.3390/en13236306_

Round 1
Reviewer 1 Report
Please see the attached file.

Reviewer 2 Report
About the work:
The work of Polyakov et al. describes the microwave synthesis of ZIF-7 MOF material in different solvents in short times such as 15 mins. The obtained materials are well and sufficiently characterized. The results are well discussed. The findings can be interesting for MOF-related researchers since the authors provide an alternative synthetic knowledge on ZIF-7 and its phases.
However, the novelty of the paper seems to be limited to the synthetic part. The paper is titled "Optimization of ZIF-7 MW synthesis and its application for hydrogen storage" however we do not encounter a hydrogen storage section in the introduction. What is the state of the art? And, based on my brief search, the results about hydrogen storage presented here are not very impressive considering the existing literature on other MOFs. Therefore, the title should change: "H2 storage application" should be removed and the paper should be more of a "synthetic method" oriented paper. Or at least, it can be smoothened such as "potential candidate for H2 storage material" or something in this sense.
About the experiments:
- The authors should try other methods to activate the so-called ZIF-7 DMF and present new data for porosity. 60C under vacuum seems insufficient to me.
- The authors should explain more the reasons of the two populations of particle size in the so-called ZIF-7 DMF. Are we sure that they are both ZIF-7? Or do we have a non-crystalline phase too?
About the text:
The abbreviations should all first appear in their full names. We don't see the full names of ZIF-7, MW, DMF etc... I have also noticed several errors in the English. The possessive phrases should be re-visited such as "researcher's attention" in line 38 which should have been "attention of researchers" or "researchers' attention". There are many others like this.
Also, the "N2 adsorption isotherms" should be" N2 adsorption-desorption isotherms" or simply "N2 sorption isotherms". In line 135 reflection should be diffraction. In line 227 and 230: ZID-7 instead of ZIF-7? The text requires a major checking before publishing.
Round 2
Reviewer 2 Report
I can now advice this manuscript for publication after the following text corrections that I could detect:
- The abbreviations should first appear in parenthesis, then can be used throughout the text. For example, N,N-diethylformamide doesn't need to appear several times in the abstract in its full name.
- line 12 "ZIF-7 (Zeolitic imidazolate framework)" should be written as zeolitic imidazolate framework-7 (ZIF-7)
- line 40 "Metal-organic framework" should be written as "metal-organic framework". No capital letters needed.
- line 43 "Flexible MOFs"
- line 221 please reduce the font size of that sentence.
- line 238 "will be completely unavailable"
- line 257 N2 sorption
